# OpenAWSEM with Open3SPN2: A fast, flexible, and accessible framework for large-scale coarse-grained biomolecular simulations

Wei Lu[1,3], Carlos Bueno[1,2], Nicholas P. Schafer[1,2,5], Joshua Moller[6,7], Shikai Jin[1,4], Xun Chen[1,2], Mingchen Chen[1], Xinyu Gu[1,2], Aram Davtyan[1], Juan J. de Pablo[6,7], Peter G. Wolynes[1,2,3,4] *

**1** Center for Theoretical Biological Physics, Rice University, Houston, Texas, United States of America, **2** Department of Chemistry, Rice University, Houston, Texas, United States of America, **3** Department of Physics, Rice University, Houston, Texas, United States of America, **4** Department of Biosciences, Rice University, Houston, Texas, United States of America, **5** Schafer Science, LLC, Houston, Texas United States of America, **6** Pritzker School of Molecular Engineering, University of Chicago, Chicago, Illinois, United States of America, **7** Argonne National Laboratory, Lemont, Illinois, United States of America

* pwolynes@rice.edu

**Data Availability Statement:** All relevant data are within the manuscript and its Supporting information files. All codes can be found in GitHub:

## Abstract

We present OpenAWSEM and Open3SPN2, new cross-compatible implementations of coarse-grained models for protein (AWSEM) and DNA (3SPN2) molecular dynamics simulations within the OpenMM framework. These new implementations retain the chemical accuracy and intrinsic efficiency of the original models while adding GPU acceleration and the ease of forcefield modification provided by OpenMM's Custom Forces software framework. By utilizing GPUs, we achieve around a 30-fold speedup in protein and protein-DNA simulations over the existing LAMMPS-based implementations running on a single CPU core. We showcase the benefits of OpenMM's Custom Forces framework by devising and implementing two new potentials that allow us to address important aspects of protein folding and structure prediction and by testing the ability of the combined OpenAWSEM and Open3SPN2 to model protein-DNA binding. The first potential is used to describe the changes in effective interactions that occur as a protein becomes partially buried in a membrane. We also introduced an interaction to describe proteins with multiple disulfide bonds. Using simple pairwise disulfide bonding terms results in unphysical clustering of cysteine residues, posing a problem when simulating the folding of proteins with many cysteines. We now can computationally reproduce Anfinsen's early Nobel prize winning experiments by using OpenMM's Custom Forces framework to introduce a multi-body disulfide bonding term that prevents unphysical clustering. Our protein-DNA simulations show that the binding landscape is funneled towards structures that are quite similar to those found using experiments. In summary, this paper provides a simulation tool for the molecular biophysics community that is both easy to use and sufficiently efficient to simulate large proteins and large protein-DNA systems that are central to many cellular processes. These codes should facilitate the interplay between molecular simulations and cellular studies, which have been

https://github.com/npschafer/openawsem, and
Open3SPN2, and https://github.com/cabb99/
open3spn2.

**Funding:** WL, CB, NPS, SJ, XC, MC, XG, AD and
PGW were supported by the Center for Theoretical
Biological Physics and sponsored by an NSF grant
(PHY- 2019745) and by the D. R. Bullard-Welch
Chair at Rice University, Grant C-0016. JM and JJP
were supported by NSF grant BIO/MCB 1818328.
The funders had no role in study design, data
collection and analysis, decision to publish, or
preparation of the manuscript.

**Competing interests:** The authors have declared
that no competing interests exist.

hampered by the large mismatch between the time and length scales accessible to molecular simulations and those relevant to cell biology.

## Author summary

The cell's most important pieces of machinery are large complexes of proteins often along with nucleic acids. From the ribosome, to CRISPR-Cas9, to transcription factors and DNA-wrangling proteins like the SMC-Kleisins, these complexes allow organisms to replicate and enable cells to respond to environmental cues. Computer simulation is a key technology that can be used to connect physical theories with biological reality. Unfortunately, the time and length scales accessible to molecular simulation have not kept pace with our ambition to study the cell's molecular factories. Many simulation codes also unfortunately remain effectively locked away from the user community who need to modify them as more of the underlying physics is learned. In this paper, we present OpenAWSEM and Open3SPN2, two new easy-to-use and easy to modify implementations of efficient and accurate coarse-grained protein and DNA simulation forcefields that can now be run hundreds of times faster than before, thereby making studies of large biomolecular machines more facile.

This is a *PLOS Computational Biology* Software paper.

## Introduction

In recent decades, experimental methods for studying biological systems have made great strides providing dynamic and structural information across a range of scales. Nevertheless, most experimental probes are still very indirect, with a wide gap between what can be measured directly and what scientists actually want to understand and visualize. Modern theoretical frameworks for organizing our thinking along with computational simulation codes begin to allow the detailed mechanisms of biomolecular assemblies to be laid bare. The development of physical simulation models allows mechanistic ideas that are often only inferred indirectly from structural biology to be tested rigorously in a quantitative way rather than remaining attractive but qualitative hypotheses. Biomolecular simulations, in fact, are now beginning to uncover previously unforeseen mechanisms on the molecular level.

When writing down a mathematical description of the forces acting on biomolecules, an important first decision to make is what degree of detail is needed to represent the relevant motions of the biomolecules within their environment. In particular, one must decide which of the atomic degrees of freedom should be kept and which can be averaged over. Retaining all of the atomic degrees of freedom gives rise to the popular all-atom models of biomolecules immersed in a solvent which is also described in atomic detail. While these models are computationally costly to simulate, they can be quite accurate and have recently been used successfully to fold small proteins and even now begin to allow study of the dynamics of larger systems. [1, 2] The great amount of detail in the all-atom representation often leads us to forget that all-atom models today still make physical assumptions like the additivity of the

intermolecular forces, which may not be fully accurate in all situations. Averaging over the solvent degrees of freedom yields tremendous computational cost savings. The gain in efficiency arises from two factors: first, when we simulate a solvated biomolecule in full atomic detail, the vast majority of the atoms belong to the solvent. Eliminating them from detailed consideration then greatly reduces the number of computational operations needed to follow the dynamics. Second, as parts of the biomolecule move through the solvent they are constantly buffeted by collisions with the nearby solvent molecules. These collisions dramatically slow down the large scale motions that usually are of the most interest, yet in the main these frictional effects do not change the structural character of the motions.

Averaging over all of the solvent degrees of freedom while retaining a fully atomically detailed representation of the biomolecule thus already yields significant computational advantages. While solvent averaging alone increases computational efficiency, additional computational savings can be had by simplifying the representation of the biomolecule itself. Here again, there are two ways computational time is saved. First, there is a direct savings related to the need to compute a still smaller number of forces. Second, one can choose to intentionally speed up certain internal motions that are otherwise slow in a typical all-atom model by lowering torsional barriers, such as the rotation of backbone Ramachandran dihedral angles. Opting for a coarse-grained representation of a biomolecule, by facilitating sampling, greatly expands the number of biological questions that can be effectively studied.

While it is convenient to average over the solvent and detailed side chain degrees of freedom, the thermodynamic effects of the solvent and the side chains are subtle—considerably more subtle than the buried surface area model. In proteins, it is well known that bulk aqueous solvent gives rise to an effective hydrophobic attraction between non-polar residues. [3] This effect motivated the buried surface area approximation. It is less widely known that specifically bound water molecules also mediate interactions between pairs of polar residues; these give rise to an effective hydrophilic interaction. [4, 5] These water-mediated interactions are quite important in protein complexes. One efficient way of handling such phenomena is to alias such interactions back onto the protein degrees of freedom. Doing this leads to strongly non-additive forces. It is commonly believed that averaging over any of the degrees of freedom lowers the reliability of a model. For biomolecules, however, the all-atom force fields have themselves generally been parameterized by experimental data just as the coarse grained models are. The greater freedom of formulating coarse grained models however has long encouraged the use of machine learning strategies to determine these parameters. Such machine learning increases the accuracy of the description. [6] The resulting sophisticated coarse grained models have proved surprisingly effective in describing biomolecular dynamics both in folding and function, even in a quantitative sense. [7]

## Design and implementation

The coarse-grained protein folding force field known as the Associative memory, water-mediated, structure and energy model (AWSEM) is the latest iteration of a series of coarse-grained models that have been primarily developed in the Wolynes and Papoian groups over the last several decades [8]. AWSEM employs a detailed backbone representation along with a single interaction site for each side chain. The AWSEM force field includes an implicit solvent model with a hydrophobic burial term along with explicit water-mediated nonadditive interactions between the residues. AWSEM-MD is an implementation of the AWSEM model in the LAMMPS molecular dynamics package [9]. AWSEM-MD has been successful in predicting the structures of globular $\alpha$-helical proteins [8], both designed and natural $\alpha/\beta$ proteins [10], and polytopic $\alpha$-helical membrane proteins [11]. AWSEM-MD has also been used to study

protein association [12] and aggregation [13]. Recently, AWSEM-MD has been used to predict the folds of large proteins by incorporating co-evolutionary information [14] and 3D template information [15]. It has also performed quite well in recent CASP competitions. [16]

Nucleic acids are important partners with proteins in biology and it is desirable to study their dynamics with compatible computational tools. 3SPN.2 is a Coarse Grained DNA model developed by the de Pablo group that models the DNA molecule using 3-sites-per-nucleotide: a particle for the phosphate group, a particle for the sugar and a particle for the nucleobase [17]. 3SPN.2 provides a flexible representation for the DNA backbone, and employs a detailed representation of the base pairing interaction and DNA electrostatics. 3SPN.2C also describes the DNA sequence dependent curvature [18]. 3SPN.2C has already been used in combination with AWSEM to study protein-DNA complexes, such as the nucleosome [19] and NF-$\kappa$B DNA complexes [20].

As the force fields that are used to model protein and protein-DNA systems become more complicated, and as the systems being studied become larger, the software used to model these systems must also evolve. The challenges are clear: for example, in a recent study of chromosome organization proteins [21], AWSEM combined with co-evolutionary information was used to study a protein complex having a total of 3964 residues. For these large systems, even relatively short simulation runs of 100 ns laboratory time took up to 24 hours to obtain using LAMMPS code. In the present paper, we will show how the OpenMM framework can be used to speed up such simulations using GPUs and how OpenMM framework allows one to introduce novel interactions in the simulation force field models with relative ease.

The LAMMPS simulation package employs a parallelization scheme that is based on spatial-decomposition, with each CPU handling a separate contiguous region of space. Information about the forces that act across the boundaries of these domains is passed between the processors at each timestep. This parallelization scheme is relatively simple to implement due to its nearly universal structure with respect to different forcefields. This approach to parallelization scales very well for simulations of bulk liquids and solids, where the system has a nearly uniform density. For simulations of biomolecules with an implicit solvent forcefield, like AWSEM and 3SPN2, however, spatial decomposition can be inefficient because the systems have highly heterogeneous local densities. Processors that compute the interactions inside of the mostly empty boxes will ordinarily then be idle while waiting for the processors that compute the interactions inside of those boxes that are full of atoms. A spatial-decomposition scheme that dynamically adjusts the sizes of the CPU-domains can only partially compensate for this effect. For implicit solvent models, the force-based parallelization scheme employed by OpenMM turns out to be much more efficient, especially when implemented on GPUs. [22] OpenMM was developed with the intention of being compatible with multiple hardware platforms including GPUs. It provides a high level application programming interface (API) that removes the burden of writing platform specific codes. Traditionally, computational scientists have designed forcefields for single CPUs and then only later would spend time modifying their codes to support simulations on multiple CPUs and even more time on adding GPU support. With OpenMM, one only needs to write down the equations describing the forcefields once, and the software automatically compiles optimized code that can be run on all platforms including a single CPU, multiple CPUs, and GPUs (with both CUDA and OpenCL support).

OpenMM provides various flexible custom force templates to ease the implementation of forcefields with new functional forms. To implement OpenAWSEM and Open3SPN2, we used the custom force template that best fits each term in the Hamiltonians. For example, the "CustomNonbondedForce" is the best choice for the excluded volume term, which acts between every pair of atoms, while the "CustomBondForce" supports a very wide range of functional forms and is appropriate for terms that involve only a small subset of the system's atoms.

Another interesting situation that OpenMM flexibly encodes is AWSEM's water-mediated interaction. Since the water-mediated interactions depend on the local density around each interacting atom, the local density around each residue has to be computed first before computing the mediated interactions. This two stage feature can be implemented using the "CustomGBForce" template, which was originally intended to support another two stage energy term: the Generalized Born-type potentials.

The custom force templates allow for rapid prototyping of new potential terms. For each new potential, only the energy formula needs to be specified, while its derivatives are automatically computed for the purposes of computing the forces. By automating the derivative calculation, even non-experts can design and implement new force fields readily. In this paper, we will illustrate this capability of the OpenMM framework by introducing two new features into AWSEM. The first new feature is a contact term that depends on the degree of burial of a residue in a biological membrane. This energy can be used to describe proteins that have both cytoplasmic parts that are surrounded by water, and other parts that are buried in a membrane, which are thus surrounded by lipid primarily. The second new nonadditive potential we introduce and explore is a many-body disulfide bond term that prevents the unphysical clustering of Cysteines that can occur when disulfide bonds are modeled using a naïve pair potential that must per force be very strong. This potential allows us to recapitulate the early experiments of Anfinsen on ribonuclease that started the experimental study of protein folding mechanism. [23, 24]

## Results

### Benchmark 1: Protein-only simulations

When AWSEM was first implemented using LAMMPS 8 years ago, dynamic studies of proteins mostly focused on proteins having less than a thousand residues. This limited focus was due both to the computational cost of studying larger system, and partly, to the scarcity of experimentally solved structures of large biological machines. The structures of larger proteins and their complexes are now being obtained at an unprecedented pace, thanks especially to the development of Cryo-EM structure determination methods. One recently solved large protein, gamma secretase has drawn lots of attention due to its role in Alzheimer' disease. Gamma secretase contains 1542 residues. [25] Fig 1 shows comparative benchmark results for

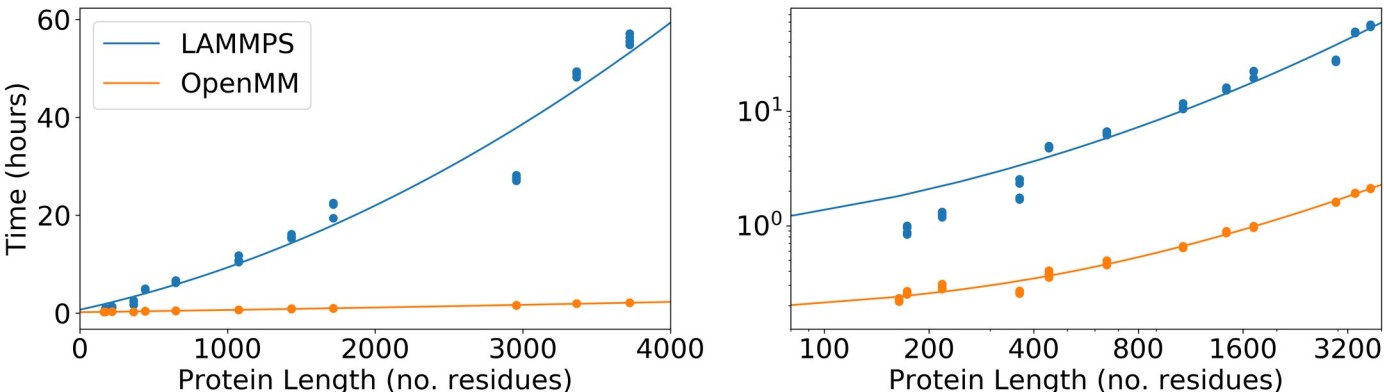

**Fig 1. Benchmark timing results for AWSEM simulations with the LAMMPS and the OpenMM implementations on a linear scale (left) and on a log scale (right).** The x-axis is the number of residues in the proteins that are being simulated. The y-axis shows the number of computer hours needed to run a 1 million-step simulation. Each protein was simulated 5 times using each implementation. The lines are quadratic fits. The simulation protein set was chosen to have a wide range of protein sequence lengths ranging from 164 residues to 3724 residues.

simulations using OpenAWSEM and using LAMMPS for proteins with various lengths. For a protein with 3724 residues (PDBid: 4qqw), a simulation of 4 million steps corresponding roughly to 20$\mu s$ in laboratory time took more than 200 hours (8 days) using LAMMPS. The same simulation takes only about 8 hour using OpenAWSEM, thus making millisecond simulations feasible within a few days.

## Benchmark 2: DNA-only simulations

To test the scaling of the runtime of Open3SPN2 for nucleic acids, we ran several random DNA sequences of different lengths using the 3SPN2.C forcefield. The DNA strands were simulated using LAMMPS and using OpenMM for 1 hour and, from these test runs, we estimated the time needed to run 1 million steps. As shown in Fig 2, the OpenMM implementation of 3SPN2.C reduces the simulation time of long DNA strands ranging in size from 250 bp up to 1.5kb DNA strands. For short sequences, the GPU is underutilized and the greater overhead associated with using the GPU results in longer overall simulation times. For the 1.5 kb case, we found a fourfold improvement in simulation speed. For longer DNA strands, the speedup will be greater due to better scaling. This improvement in the simulation speed allows the study of DNA dynamics on much longer timescales even for more complex systems such as DNA origamis or small sections of chromosomes.

## Benchmark 3: Protein-DNA simulations

To assess the speedup of DNA-protein simulations we selected several protein-DNA complexes that have a diverse range of lengths for both the protein and the DNA sequences. We included in this test set only structures from the PDB that contained a single protein chain and a single DNA chain. We simulated each complex 5 times for 1 hour using each implementation and estimated how much time would be required to run 1 million steps. Fig 3 shows an improvement of the simulation speed of protein-DNA complexes by 1 to 2 orders of magnitude. The largest structure that we simulated was RecA, a protein with 2050 amino acids, in complex with a 18 nucleotides ssDNA (PDBid: 3cmu). In this case, we obtained a 300-fold speedup.

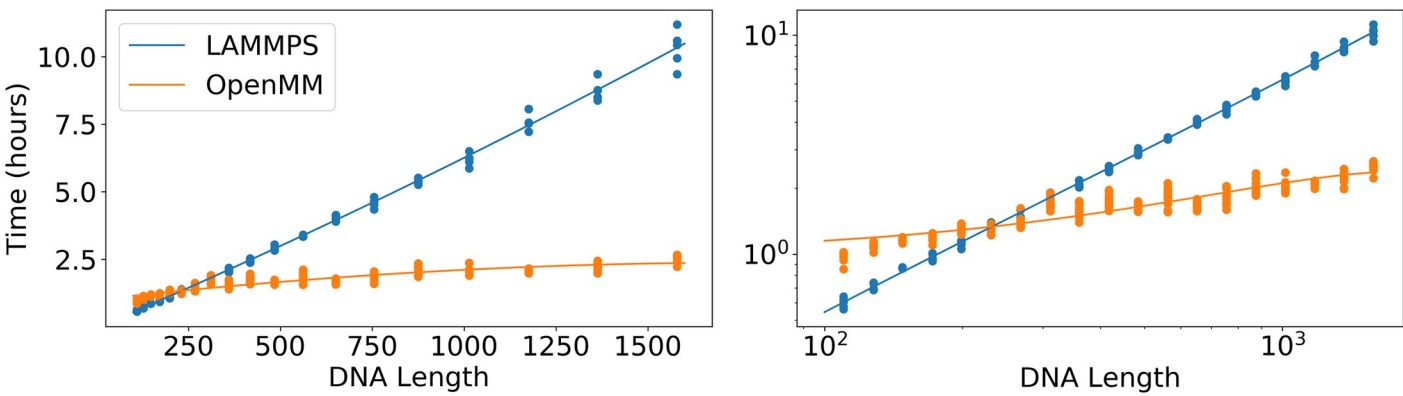

**Fig 2. Benchmark timing results for 3SPN2 simulations with the LAMMPS implementation of 3SPN2 and the OpenMM implementation of 3SPN2 on a linear scale (left) and on a log scale (right).** The x-axis is the number of nucleotides in the DNA that is being simulated. The y-axis shows the number of computer hours that are needed to run a 1 million-timestep simulation. Each DNA length was simulated 5 times using each implementation. The lines are quadratic fits. The DNA lengths range from 110 nucleotides to 1580 nucleotides.

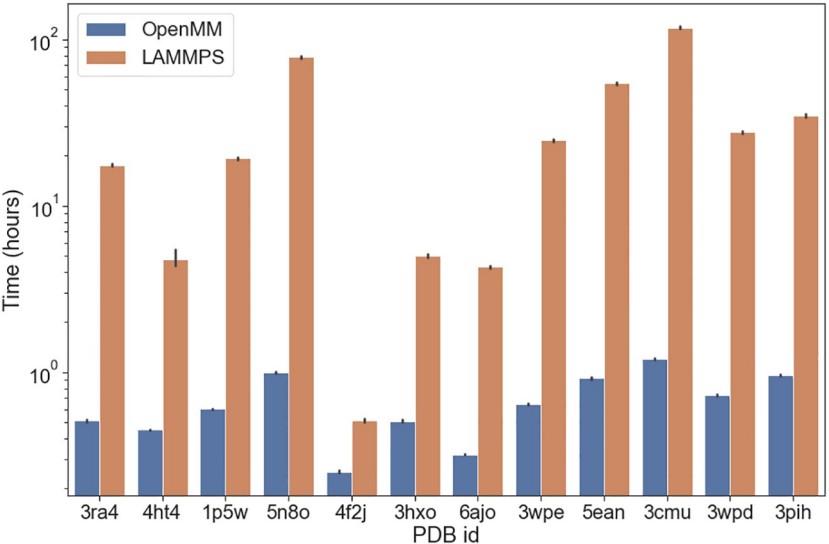

**Fig 3. Benchmark results for AWSEM-3SPN2 simulations of protein-DNA complexes using the LAMMPS and the OpenMM implementations of both forcefields on a linear scale (left) and on a log scale (right).** The x-axis shows the PDB ID. The y-axis shows the computer hours needed to simulate for 1 million steps. Each complex was simulated 5 times using each implementation. The protein length ranges from 52 nucleotides to 2050 amino acids, while the DNA length ranges from 2 to 40 nucleotides.

## Application 1: Protein-DNA interface prediction

As an example of simulating protein-DNA interactions, we characterized the capability of the AWSEM-3SPN2 Hamiltonian to predict the correct protein-DNA binding interface of the sporulation specific transcription factor Ndt80 (PDBid: 1mnn). At a constant temperature of 300K, the protein and DNA in the crystal structure were first pulled $100\mathring{A}$ apart and run for 2.5 million steps; following this, a weak, non-specific force was used to pull them back together while being run for another 2.5 millions steps. Following this, the pulling force was released and the complex was simulated for another 5 million steps to let it relax. To reduce the effects of binding to only a short length of DNA, we extended the crystallized DNA by adding DNA made with 100 A/T base pairs to both ends of the double stranded DNA using the 3DNA package [26].

The OpenAWSEM-Open3SPN2 cross-interaction is given by electrostatic interactions between the DNA phosphates and charged residues of the protein, as well as excluded volume terms. The current implementation lacks specific interactions that depend on the nucleotide type and amino acid type. Therefore, would it not through indirect DNA conformation-mediated effects, the protein would not be expected to prefer binding to any particular stretch of nucleotides on the DNA. The part of the protein surface that binds to the DNA and the orientation of the bound protein with respect to the DNA, however, is somewhat specific. To evaluate the quality of the DNA-protein interface, while focusing on finding the native binding pocket of the protein, we quantified the quality of the docking in terms of the number of contacts that the protein makes with any location along the DNA. A residue in the protein is said to make such a "symmetrized" contact with DNA when the $C_\beta$ atom in the residue is closer than 1.8 nm to a Phosphate of DNA in the crystal structure and where also, in the predicted structure, this Cb atom is found within 1.8nm of a Phosphate of the DNA. For PDB ID 1mnn, there are 135 such native contacts. The interface energy is defined as the sum of protein-DNA excluded volume energy and the electrostatic interaction energy between the protein and the

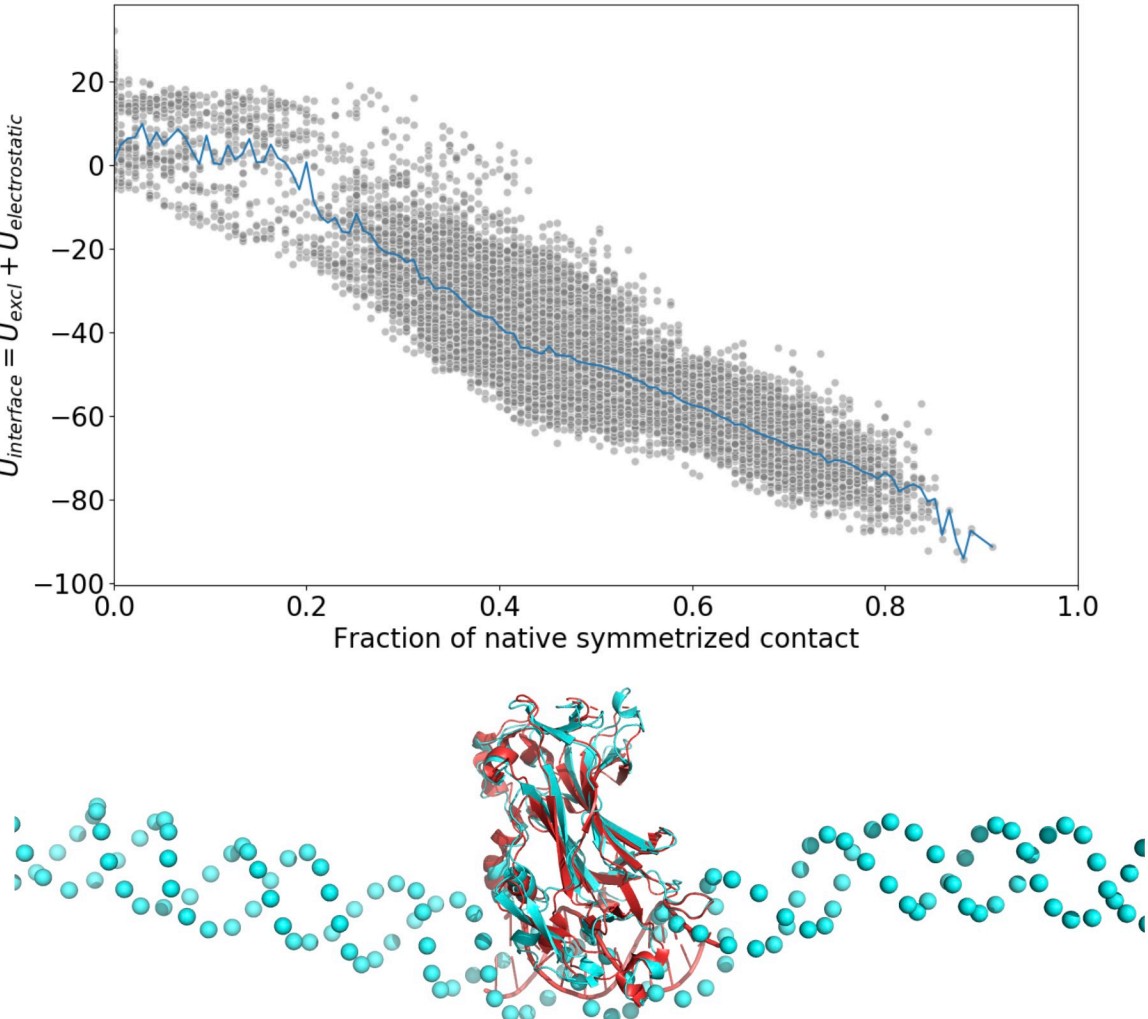

**Fig 4. A scatter plot of the interaction energy between the DNA and the protein versus the fraction of the symmetrized native contacts formed at each time frame during the last 7.5 million steps of simulations from 10 runs.** The average energy as a function of the number of symmetrized native contacts is indicated with blue line. A simulation snapshot showing the overlap of the crystal structure (colored in red) and the predicted structure (colored in cyan) that has the lowest interface energy. There is a high correlation between the protein-DNA interface energy and the number of symmetrized contacts, indicating that the binding process is funneled to the correct interface. The overlap figure was created by aligning only the protein parts of the crystal structure and the predicted structure. We see that the DNA in both structures turns out to be aligned quite well, showing good structural agreement between the lowest energy simulated structure and the experimental structure.

DNA. As can be seen in Fig 4, there is a strong correlation between the protein-DNA interface energy and the quality of the protein-DNA interface, and the orientation of bound protein relative to the DNA matches that found by experiment.

## Application 2: Potentials that depend on locations of residues relative to a membrane

The water-mediated potential introduced by Papoian et al. [8] acknowledged that residues interact not only when they are directly in contact but also when they perturb the surrounding water, which in turn changes the energetics of more distant residues. The parameters for this potential were optimized using an energy landscape theory inspired machine learning

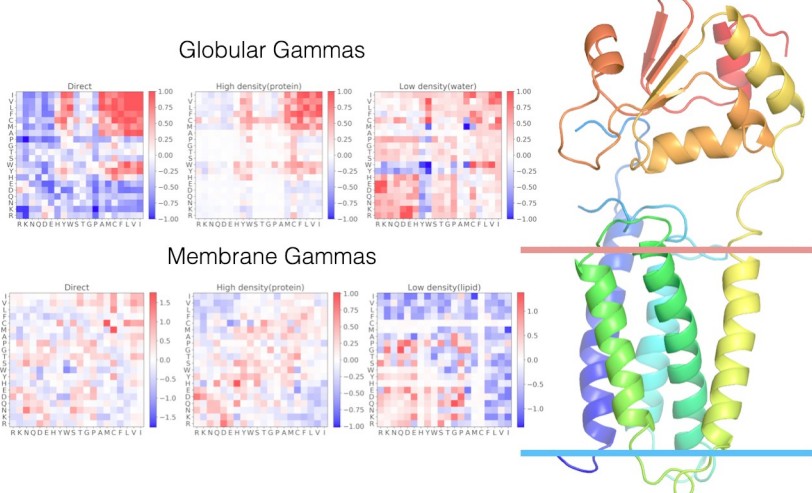

**Fig 5. A schematic figure for the Z-dependent contact potential.** The residues outside of the membrane, where the membrane boundary is indicated by the two colored lines, interact using the globular parameters. The residues inside the membrane interact using the membrane-optimized parameters. If one residue is inside, while another one is outside, the pair interacts as if they both were in water. In the heat maps on the left side of the figure, red color indicates a favorable interaction between the pair of residues indicated on the horizontal and vertical axes, whereas blue color indicates an unfavorable interaction. Separate heat maps are shown for the direct, low-density, and high-density interaction matrices in the water (globular) and membrane environments.

algorithm [7, 27–30]. Energy landscape theory provides a recipe whereby a transferable energy function can be learned by searching for the most funnel like landscape in a class of energy models. The funnel-like character of the landscape is measured by a Z score, $Z = (E_{native} - E_{mg})/\sigma(E_{mg})$. This quantity is then maximized while maintaining $E_{mg}$ constant. $E_{mg}$ is the average energy of a set of misfolded decoy structures. Using this strategy leads to an optimal set of parameters to discriminate between properly folded and misfolded structures. In the simplest model these parameters are the strengths of the interactions for different types of residue pairs at various distances and how these interactions vary with the local density of protein and by contrast with the local density of solvent water. The AWSEM potential has proved to be very successful in structure prediction and has allowed exploration of many aspects of protein functional motions [12, 13]. The water-mediated potential was originally designed for globular proteins, but the same optimization scheme was used also to find a transferable energy function that would fold membrane proteins, [11] in their membrane environment; the residue pair interactions then are mediated by lipids instead of by water. Following the same procedures as used for the globular proteins, the parameters for proteins that are found entirely inside the membrane were optimized to discriminate proper folds. Many proteins, however, have some of their parts inside the membrane while other parts of the protein remain outside in the cytoplasm. To study such systems we need a potential that can dynamically switch from being water-mediated to lipid mediated based on the position of the residues relative to the bilayer. Fig 5 shows the schematic of this potential.

Here, we introduce a z-dependent contact term that allows such dynamic switching. The interactions smoothly transition between the membrane mediated interactions and water-mediated interactions depending on the location of the interacting residues with respect to the membrane as measured by a height z. We define the new contact potential term $V_{contact}$

through the following equations:

$$V_{contact} = \sum_{j-i>9} V_{contact}(i,j) \tag{1}$$

$$V_{contact}(i,j) = (1 - \alpha_i\alpha_j)V_{water}(i,j) + k_{relative}\alpha_i\alpha_j V_{membrane}(i,j) \tag{2}$$

$$\alpha_i = \frac{1}{2}(tanh(\eta(z_i + b)) + tanh(\eta(b - z_i))), \tag{3}$$

where $b = 1.5nm$, $\eta = 10nm^{-1}$. $V_{water}(i,j)$ and $V_{membrane}(i,j)$ are the contact terms as defined in previous paper [8, 11].

Since both sets of parameters in the Hamiltonian were previously optimized without acknowledging the presence of the other terms, we also need to introduce a new parameter $k_{relative}$ that controls the relative strength of the membrane mediated and the water-mediated interactions. A high $k_{relative}$ favors forming contacts inside the membrane, while a low $k_{relative}$ favors forming contacts in water. To determine the optimal value of $k_{relative}$, we again employ the energy landscape optimization learning scheme. The decoys for implementing this scheme were generated by shifting the proteins vertically and rotating them. One then optimizes the $k_{relative}$ while keeping the previously determined parameters fixed. This machine learning scheme was employed using a test set obtained by downloading the complete Alpha-helical polytopic database, a total of 1561 proteins, from the Orientations of Proteins in Membranes (OPM) database. [31]. The advantage of the OPM database over the traditional RCSB protein data bank is that it also spatially aligns membrane proteins relative to the membrane. The training proteins must have significant parts both inside and outside the membrane. Therefore, for each protein, we computed the fraction of the residues that are found inside the membrane

$$\chi = \frac{1}{L}\sum_{i=1}^{L}(abs(z_i) < 15\mathring{A}), \tag{4}$$

where $z_i$ is the z coordinate of CA of residue i, $L$ is the protein length. For training we only kept those proteins with $\chi$ between 0.2 and 0.8. We also removed those proteins that have more than 2000 residues in order to speed up the optimization. This yielded a set of 1116 training proteins. For each protein, we then generated 240 decoys. These were generated first by rotating them along the x axis with 12 different orientation at: 0, 15, 30, 45, 60, 75, 90, 105, 120, 135, 150, 165 degrees, and then shifting the structure vertically by 20 different displacements: -40, -36, -32, -28, -24, -20, -16, -12, -8, -4, 0, 4, 8, 12, 16, 20, 24, 28, 32, 36 angstroms along the z-axis. To carry out this optimization, the total energies are evaluated using the following

equations:

$$E = k_{wat}\phi_{wat} + k_{mem}\phi_{mem} + k_{mem_{burial}}\phi_{mem_{burial}} \tag{5}$$

$$\phi_{wat} = \sum_{j-i>9}(1 - \alpha_i\alpha_j)V_{water}(i,j) \tag{6}$$

$$\phi_{mem} = \sum_{j-i>9}\alpha_i\alpha_j V_{membrane}(i,j) \tag{7}$$

$$\phi_{mem_{burial}} = \sum_i A(\sigma_i)\Theta(z_i, z_m = 15\mathring{A}) \tag{8}$$

$$\Theta(z_i, z_m) = \left\{\frac{1}{2}\tanh[k_m(z_i + z_m)] + \frac{1}{2}\tanh[k_m(z_m - z_i)]\right\} \tag{9}$$

In these expression the values of $A(\sigma_i)$ are the amino acid hydrophobicities on the octanol scale of Wimley and White. [32–35] We include $\phi_{mem_{burial}}$ here because the membrane burial term also depends on the position of protein with respect to the membrane. [36] In the machine learning algorithm thus we want to find the values of $k_{wat}$, $k_{mem}$ and $k_{mem_{burial}}$ that maximize the Z score for the correct positioning and orientations of the proteins with the membrane. Since some decoys are more similar to the native positioning than are others, we reweighted the decoys when computing the decoy averages in $\langle\phi\rangle_{mg}$

$$\langle\phi\rangle_{mg} = \frac{1}{\sum_{d=1}^{N}(1 - \theta_d)}\sum_{d=1}^{N}(1 - \theta_d)\phi_d \tag{10}$$

where $N$ is the number of decoys. For each decoy, the fraction of residues that have the same pattern of burial as the native structure is defined to be $\theta_d$. Two residues are said to have the same burial assignment when either they are both inside the membrane or they are both in the cytoplasm. $\theta_d = \frac{1}{L}\sum_{i=1}^{L}\delta_i, \delta_i = \begin{cases} 1 & \text{if } (abs(z_i^0) < 15\mathring{A}) = (abs(z_i) < 15\mathring{A}) \\ 0 & \text{if } (abs(z_i^0) < 15\mathring{A}) \neq (abs(z_i) < 15\mathring{A}) \end{cases}$, where $z_i^0(z_i)$ is the z coordinate of CA of residue i in the native(decoy) structure. The optimal values of the coefficients that maximize the Z score turn out to be 1, 3.3, 3.3 for $\phi_{wat}$, $\phi_{mem}$, $\phi_{mem_{burial}}$ respectively.

To demonstrate the effectiveness of the force field obtained in this way, we selected from the database 15 proteins that have both membrane and globular parts. The folding of membrane proteins is sometimes thought to have two stages. [37] The first stage is imagined to be the insertion of the transmembrane helices into the membrane. In vivo this process is sometimes helped by the translocon [38]. The second stage of membrane folding is then the rearrangement of the now buried helices inside the membrane. To imitate the first stage, we used PureseqTM [39] first to provide an initial idea of the topology with respect to the membrane. Based on the PureseqTM prediction result, we wrote a script to assign each residue to three different regions: cytoplasmic, membrane or extracellular. Each residue is then pulled into its preliminarily predicted region according to the resulting initial assignment using a force field that only contains the backbone terms. Then, a force is applied to the two ends of the protein while applying a strong membrane term, so that the helices become well separated but still live within the membrane. Finally, the residue type dependent membrane potential is introduced along with the contact terms and an annealing protocol of 8 million steps is followed with the

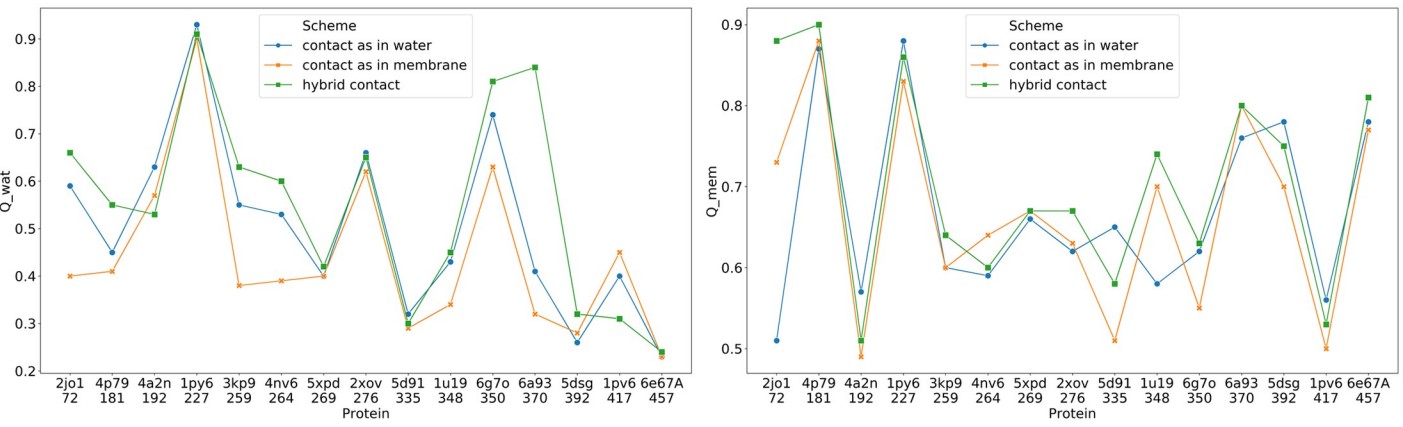

**Fig 6. Structure prediction results using the three contact potential schemes evaluated using $Q_{water}$ (left) and $Q_{mem}$ (right).** $Q_{water}$ measures the structural similarity to the native structure using only the residues that are outside of the membrane, whereas $Q_{mem}$ measures the structural similarity of the structures for those residues found inside the membrane. The closer the similarity score is to 1.0, the more native like is the prediction. The hybrid potential in general performs better than either the pure globular protein model or the pure membrane model.

temperature decreasing from 800 to 200. The results for the structure prediction runs using the z-dependent contact term are compared with the results using the original contact potential in the Fig 6.

Fig 7 shows the aligned structures of the native structures and the predicted structures using the new membrane burial depth dependent contact potential.

The AWSEM annealing yields an improved assignment of the location of the helices relative to the purely sequence based method PureseqTM that was used for initial structures. In Fig 8, we see that for 10 out of 15 proteins tested, the fraction of correctly assigned location is increased after the folding. In this test set, 3kp9, 5xpd, 1u19 now have more than 10 additional residues that take on their correct native location assignments compared to what is used initially based on the PureseqTM results.

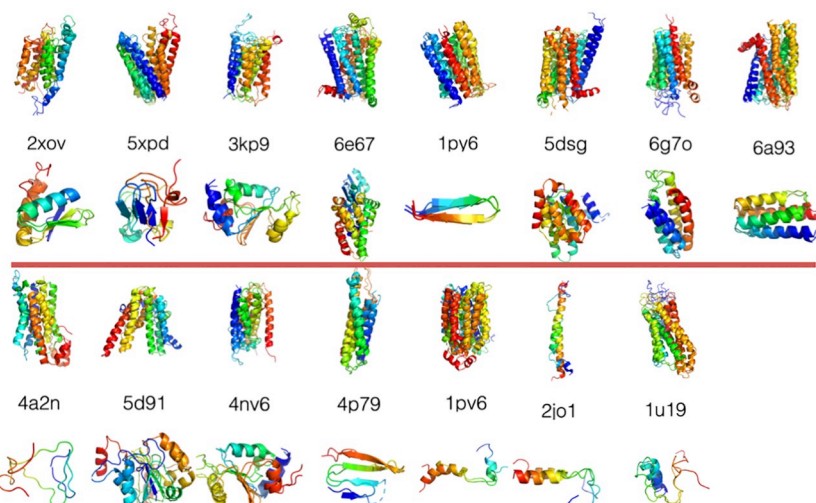

**Fig 7. Overlay of the native structures and the best $Q_{water}$ and $Q_{mem}$ structures using the membrane burial depth dependent contact potential.** For each protein, the upper figure shows the part of the protein that is found buried in the membrane and the lower part of the figure shows the globular domain.

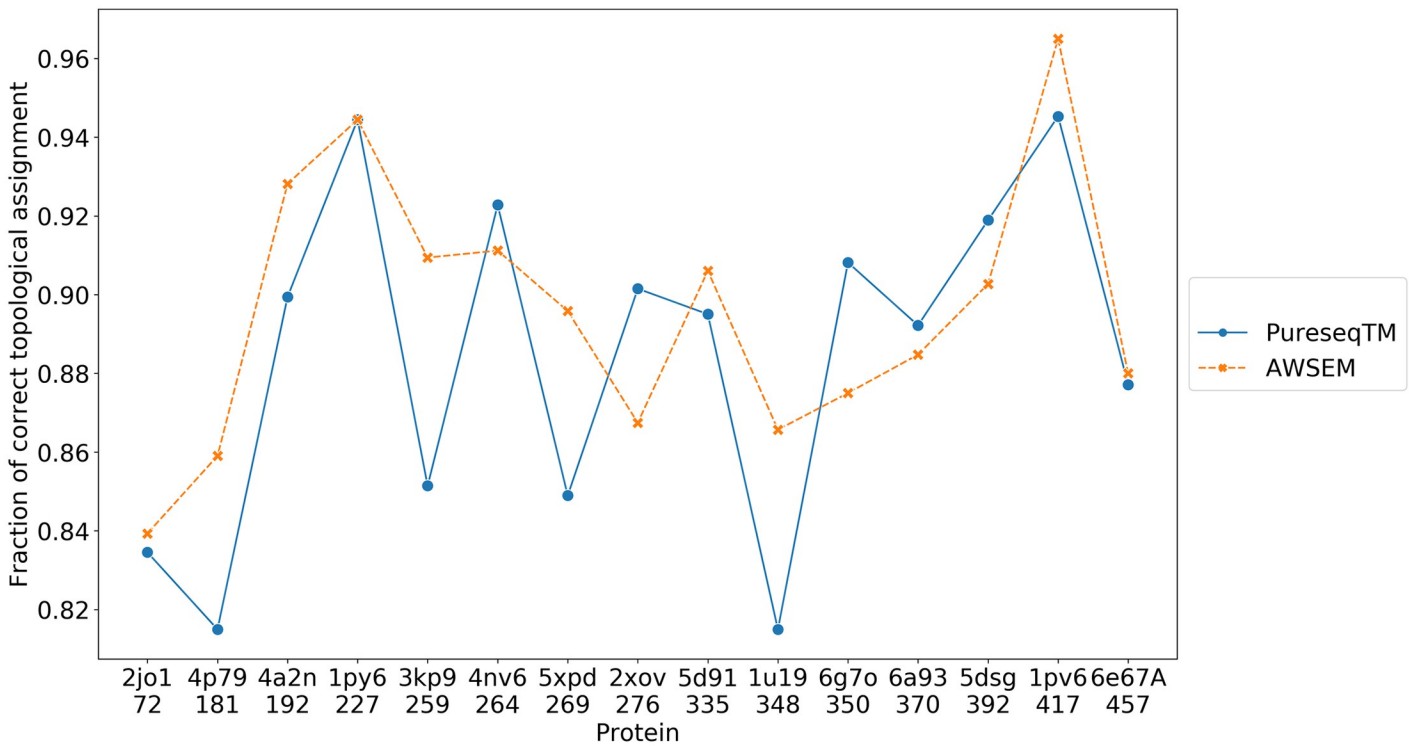

**Fig 8. The fraction of correct location assignments of the residues relative to the membrane using a purely sequence-based method (PureseqTM) and that yielded by running OpenAWSEM simulations (AWSEM).**

## Application 3: Describing many-body saturating disulfide bonds

The disulfide bond forms a very strong interaction between two Cysteines. These bonds restrain the dynamics of the protein and often control protein stability and function. Very often the smaller extracellular proteins are dominantly stabilized by a large number of disulfide linkages. If treated as a pair interaction, the strong disulfide bonds tend to condense and cluster. The covalent chemical bond, in contrast to the pair interacting potential, saturates: only one bond can be formed by each Cysteine, not more. The famous protein ribonuclease A was originally studied by Christian Anfinsen. It has four disulfide bonds. Monitoring the formation of these bonds was a key part of Anfinsen's exploration [24]. Two of the four bonds have been shown to be important for conformational stability and the other two are needed for catalytic activity. [40] Because covalent chemical bonds saturate, a simple pair-wise potential cannot model accurately Anfinsen's experiment. The saturation effect is critical: when there are only two cystines, they form a single strong disulfide bond, but when a third Cystine comes near to the two Cystines that have already formed a bond, the third Cystine shouldn't feel any strong attracting force. This is a many body effect. In this study, we tackled this saturation problem by developing a saturable many body disulfide bond interaction using the openAWSEM framework. In this potential, displayed in Eq 11, the saturation is accounted for using a density variable $\rho_i^{cys}$ that reflects the number of Cystines around residue i smoothed by a tanh function. The disulfide interaction term is then a pair interaction that is modulated by two $\rho_i^{cys}$ dependent switching functions, $\theta_{ij}^{near}$ and $\theta_{ij}^{small}$. These two switching functions are defined in Eqs 14

and 15.

$$V_{disulfide} = \sum V_{ij} \tag{11}$$

$$V_{ij} = \theta_{ij}^{near} \theta_{ij}^{small} \alpha(r_{ij}) \tag{12}$$

$$\alpha(r_{ij}) = \frac{1}{2}\left(\tanh(\kappa(r_{ij} - r_c)) - 1\right) \tag{13}$$

$$\theta_{ij}^{near} = \frac{1}{2}\left(\tanh(\kappa_s(0.2 - |\rho_i^{cys} - \rho_j^{cys}|)) + 1\right) \tag{14}$$

$$\theta_{ij}^{small} = \frac{1}{2}\left(\tanh(\kappa_s(2.2 - \rho_i^{cys} - \rho_j^{cys})) + 1\right) \tag{15}$$

$$\rho_i^{cys} = \sum_{|j-i|>1} \frac{1}{2}\left(1 - \tanh(\kappa(r - r_c))\right) \tag{16}$$

where i, j label all the Cystine residues, and $r_{ij}$ is the Cb distance between residue i and j. $\kappa_s$ is set to 20, so that $\theta_{ij}^{near}$ is 0 when the difference between $\rho_i^{cys}$ (the Cystine density around residue i) and $\rho_j^{cys}$(the Cystine density around residue j) is larger than 0.4, and $\theta_{ij}^{small}$ is 0 when the sum of those two densities is larger than 2.4. The parameters introduced to quantify the rapidity of saturation were calibrated using a database search for disulfide bonds in known crystallo-graphic structures. To determine a reasonable potential well size $\kappa$ for determining the Cystine density, our survey showed that the Cb-Cb distances between residues that form disulfide bonds fall in the range of $3.6\text{Å}$ to $4.1\text{Å}$. We therefore chose a $0.5\text{Å}$ interval over which to turn on the interaction by setting $\kappa = 10\text{Å}^{-1}$ and $r_c = 4.2\text{Å}$ in Eq 16.

To illustrate the efficiency of using the new nonadditive Cystine density dependent disul-fide bond term, we simulated the folding of ribonuclease A (1fs3), bovine pancreatic trypsin inhibitor (1bpi), alpha thrombin (1ppb) and several other cystine rich proteins selected from [41]. We tested 3 different strengths for the new potential, (k = 0, 2, 5), as well as the model that has the pairwise additive potential, which we call "original". We can see from Fig 9 that as the strength of the saturable disulfide bond term increases, the predictions become closer to the correct structure (as evaluated by the Q value). The saturable disulfide bond term signifi-cantly improves the structure prediction quality for ribonuclease A. This improvement is mainly due to the correct formation of the Cys26-Cys84 bond, which was also shown by exper-iment to be essential for protein stability. [40]

The new disulfide bond term helps specifically to form the native disulfide bonds, rather than allow the formation of mispaired Cysteines as shown in Fig 10. Even though in some cases (1tcg, 1lmm and 1ppb), the prediction quality measured by Q was not significantly affected by using the saturable disulfide interaction, the fraction of correct disulfide bonds was improved in all six proteins we tested.

When we follow the annealing trajectories for these disulfide rich proteins, we find that, consistent with the funneled nature of the energy landscape, disulfide bonds do not always form in a specific unique order, and indeed non-native disulfide bonds occasionally form and revert back to being unpaired, finally achieving a native like structure. Of course, we must bear in mind that in the laboratory this process must involve chemically tuning the oxidation of these bonds. Fig 11 shows the sequence of formation of disulfide bonds from each frame in a

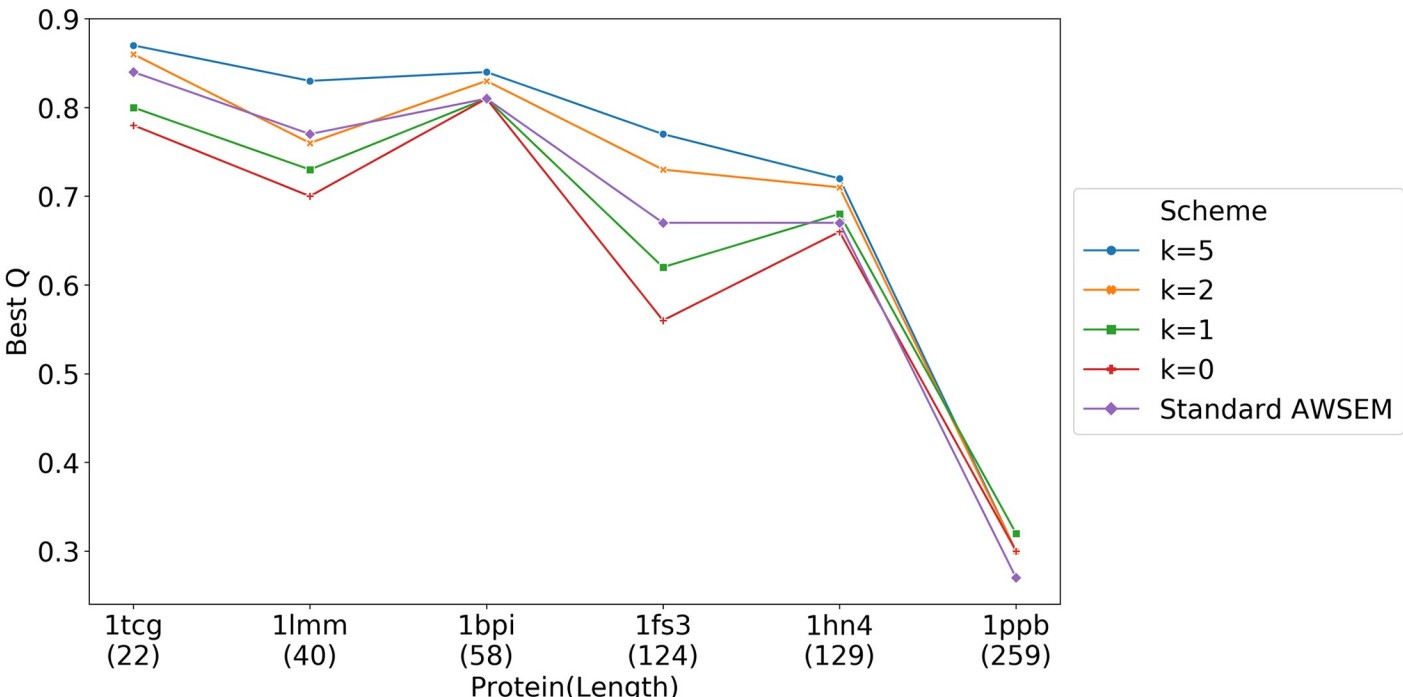

**Fig 9. Structure prediction results for six disulfide rich proteins using various strengths of the saturable disulfide bond interaction.** We plot the best Q from 20 simulated annealing runs that started from different random velocity seeds for each different value of the disulfide interaction strength. As the strength of the disulfide interactions increases, the best Q increases. 1tcg, 1lmm, 1bpi and 1ppb all have 3 disulfide bond. 1fs3 has 4 disulfide bonds, and 1hn4 has 7 disulfide bonds. The relatively modest best Q for thrombin (1ppb) probably comes from the fact that we have only modeled the main chain of the molecule, but thrombin also has a short chain that has been experimentally shown to be important for function [42].

simulated annealing trajectory of ribonuclease A. As the extended protein starts to fold from high temperature, some non-native disulfide bonds do form, but, in the end, the protein is funneled to form the correct native disulfide bonds.

As shown in Fig 12, using the standard AWSEM, only one native disulfide bond (residue 58 and residue 110) ends up being formed in most of the 20 trajectories, while the other native pairs(26-84, 40-95, 65-72) are rarely formed. In comparison, using the new Cystine density dependent disulfide bond potential, all the native pairs are finally formed.

## Discussion

We have described a new computational simulation framework for carrying out coarse grained protein-DNA simulations—OpenAWSEM and Open3SPN2. In this new framework, simulations using GPUs can achieve speedups of a factor of thirty for the simulation of proteins that have more than two thousand residues. Large lengths of DNA also can be studied more efficiently than existing CPU-based implementations. The minimal time scale for protein folding is at least microseconds [43], which indicates the size of the computational burden required to study such systems via all-atom simulations. With OpenAWSEM, folding and functional mechanisms of even very large proteins can be simulated within a reasonable amount of clock time (hours or days), thereby opening the door for a wide range of functional biomolecular dynamics studies. The codes are written entirely with Python 3, including the user interfaces. The computationally costly part of the simulations is handled by the OpenMM library, which was coded with efficiency in mind. Python 3 provides great code readability and modification efficiency, and since the codes are automatically compiled while running, the time spent in

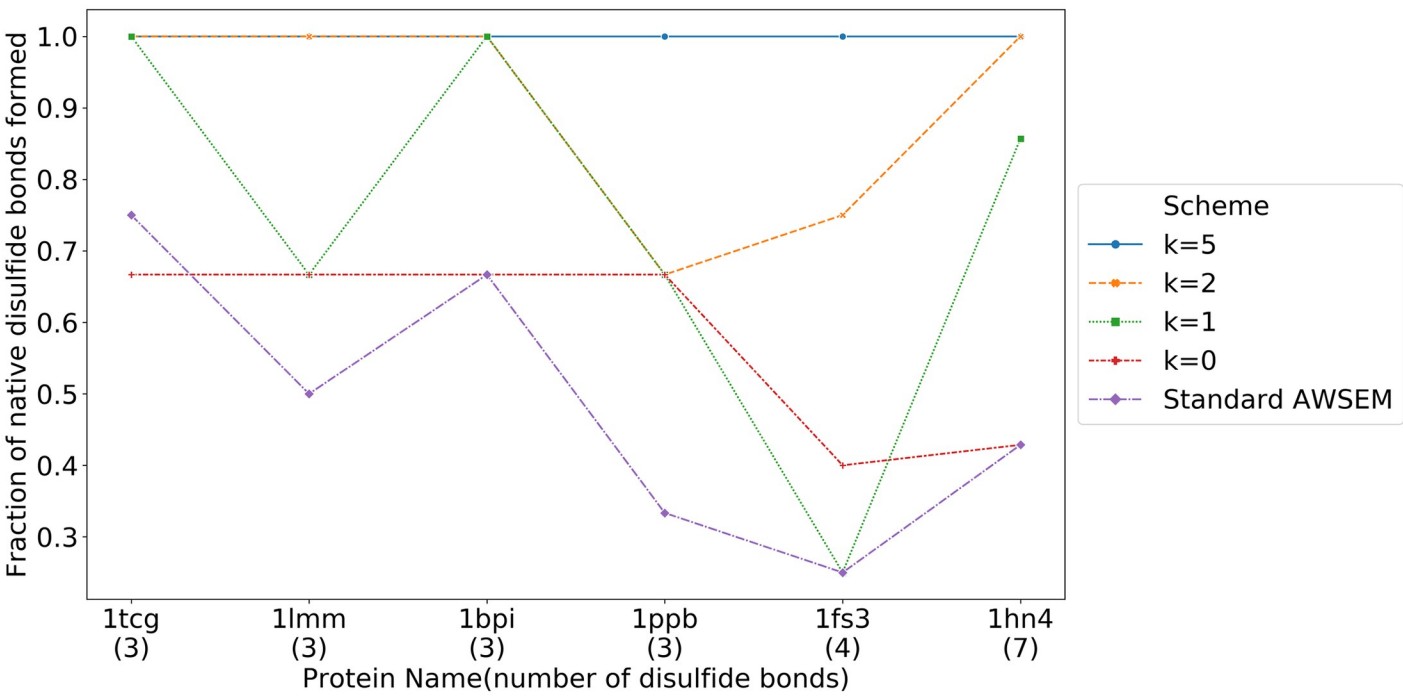

**Fig 10. The fractions of correct disulfide bonds in the predictions of several disulfide rich proteins.** These fractions are shown for several different strengths of the saturable interaction. At full strength, nearly all the pairs form correctly.

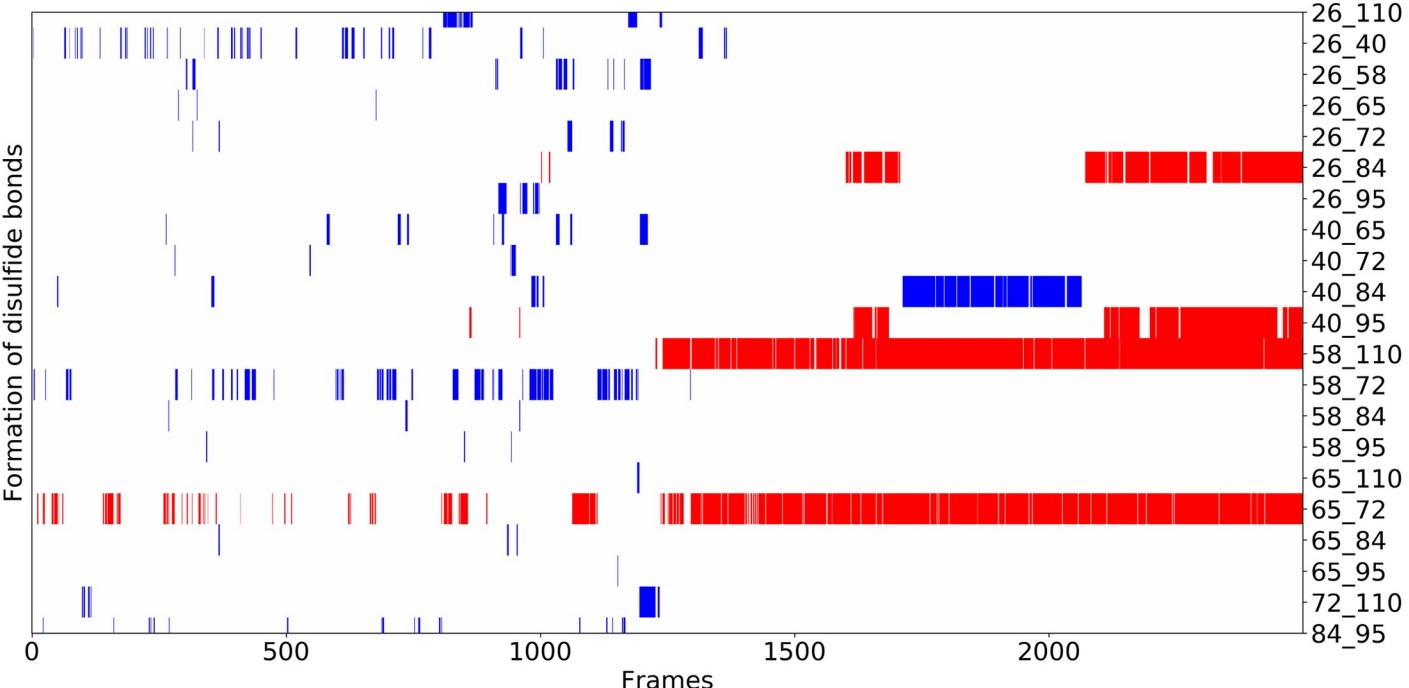

**Fig 11. The formation of disulfide bonds in a single annealing trajectory with k = 5.** Following the trajectory in time, disulfide pairs are darkened in when they are formed. Red indicates that a native disulfide bond has been formed. Blue indicates that a non-native disulfide bond has formed. The alignment of the best Q structure from this trajectory with the crystal structure is shown in SI. Its Q value is 0.77.

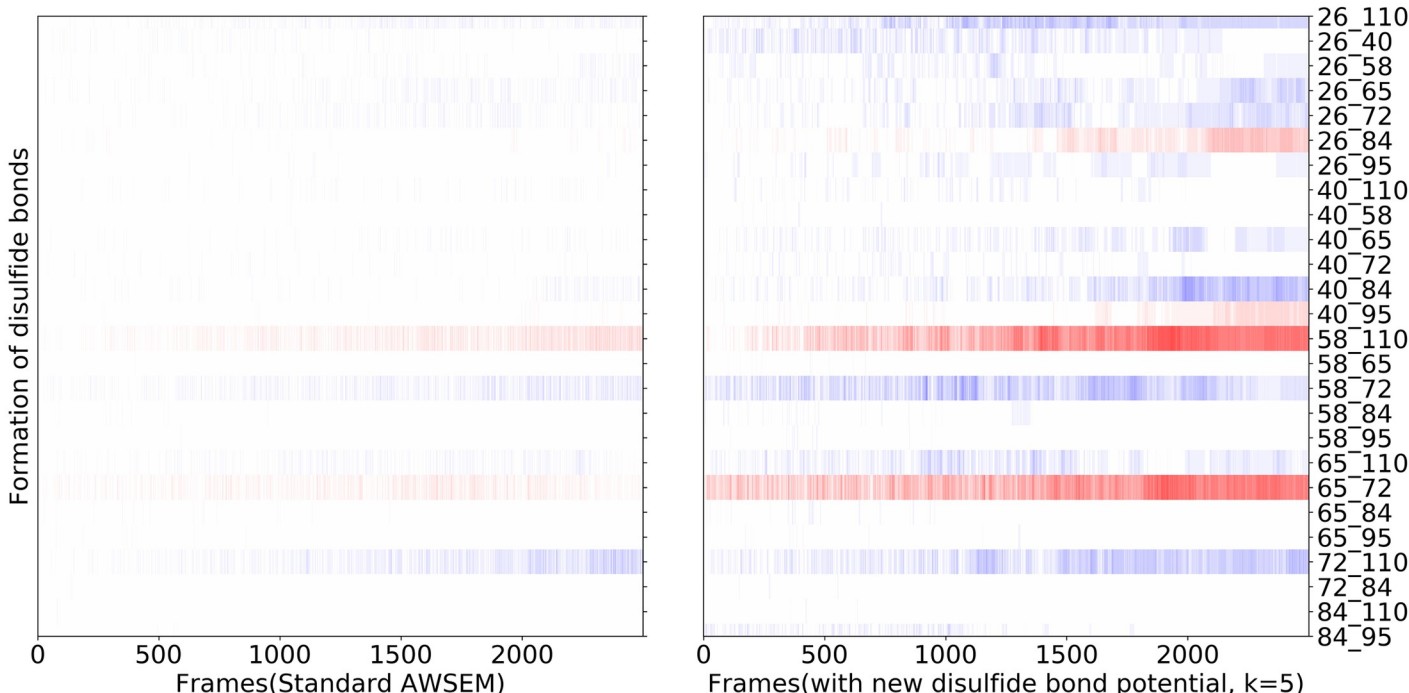

**Fig 12. The average formation of disulfide bonds as a function of time over the 20 annealing runs, with the patterns from the standard AWSEM shown on the left and patterns from the nonadditive disulfide potential runs with k = 5 shown on the right.** Red indicates that native disulfide bond has formed. Blue indicates the formation of a non-native disulfide bond. The darker the color, the larger fraction of the trajectories that form this disulfide bond during this time frame. We see that, occasionally, even with the full strength saturable interactions, sometimes non-native disulfides persist after the rapid annealings.

compilation of the code is eliminated. Also, using the automatic computation of the derivatives of the Hamiltonian instead of explicitly coding the forces greatly simplifies the introduction and implementation of new energy terms to accommodate new physics. To illustrate this feature of OpenAWSEM, we have designed and implemented two sophisticated potentials for some specialized folding situations. One of these involves the introduction of a membrane burial dependent contact potential to describe proteins that are only partially buried in membranes. We have demonstrated that using this potential for structure prediction leads to more accurate structures than when the proteins are treated as uniformly living in one environment or the other. Another energy term that was easy to code was a density dependent disulfide bonding potential that mimicks the saturation of chemical bonds. Introducing this term generally improved structure predictions and also allowed us to computationally recapitulate Anfinsen's Nobel prize winning experiments on ribonuclease. These two new potentials serve to illustrate the flexibility and extendability of the OpenAWSEM framework, and will encourage the design of future coarse grained force fields for large biomolecular simulations using this computational software framework.

## Materials and methods

### Simulation setup

The default values of the parameters in the annealing protocol for all the simulations performed in this study are given below. We maintained those values as being consistent with those typically used in the LAMMPS implementation of AWSEM-MD. (listed in S1 Document) We point out that for many problems involving very large systems, these run

parameters should be revised for optimal efficiency. As a default in the structure prediction runs, we used the langevin integrator with friction of $1ps^{-1}$, time steps of nominal $5fs$, and temperature going from 800K to 200K during simulated annealing. The simulations were carried out for 8 million steps. This corresponds roughly to 40 $\mu s$ of laboratory time. Default forces included in our study are the connectivity, chain, chi, exclude volume, rama, rama modulated by proline, rama modulated by secondary structure input file "ssweight", contact, beta, pap and fragment memory terms. Each term can be turned on and off and vary in strength and setting in the `force_setups.py` file. All OpenAWSEM and Open3SPN2 simulations were carried out with Nvidia V100 and all LAMMPS version simulations were carried out with Intel Xeon CPU E5-2650 v2 on the Rice NOTS server.

### Q-value definition

The Q-value is a measure of how similar a predicted structure is to the correct native structure. To evaluate the quality of the protein predictions we used the Q value which is defined as:

$$Q = \frac{2}{(N-2)(N-3)} \sum_{i<j-2} e^{-\frac{(r_{ij}-r_{ij}^N)^2}{2\sigma_{ij}^2}} \tag{17}$$

where N is the total number of residues, $i$ and $j$ are sequence positions, $r_{ij}$ is the distance between the CA of residue i and the CA of residue j. $r_{ij}^N$ is the distance between CA of residue i and CA of residue j in native structure, $\sigma_{ij} = (1 + |i-j|^{0.15})\text{Å}$. For $Q_{water}$, N is the number of residues outside of the membrane, and the sum is taken over all of those residues. For $Q_{membrane}$, N is the number of residues outside the membrane, and $\sigma_{ij} = 2(1 + |i-j|^{0.15})\text{Å}$

### Availability and future directions

OpenAWSEM is available at https://github.com/npschafer/openawsem website, and Open3SPN2 is available at https://github.com/cabb99/open3spn2 website. We plan to study protein-protein interactions such as the dimerization or oligomerization of membrane protein in the future.

### Supporting information

**S1 Document. OpenAWSEM and Open3SPN2 force field description.** Detailed description of the various term in openAWSEM and Open3SPN2 models along with all the parameter values.
(PDF)

### Acknowledgments

The authors would like to thank Dr. Peter Eastman the developer of OpenMM for helpful discussions during the development of the OpenAWSEM and Open3SPN2 package.

### Author Contributions

**Conceptualization:** Wei Lu, Carlos Bueno, Nicholas P. Schafer, Joshua Moller, Juan J. de Pablo, Peter G. Wolynes.

**Data curation:** Wei Lu, Carlos Bueno, Shikai Jin, Mingchen Chen, Xinyu Gu.

**Formal analysis:** Wei Lu, Carlos Bueno, Xun Chen.

**Funding acquisition:** Juan J. de Pablo, Peter G. Wolynes.

**Investigation:** Wei Lu, Carlos Bueno, Nicholas P. Schafer, Shikai Jin, Juan J. de Pablo, Peter G. Wolynes.

**Methodology:** Wei Lu, Carlos Bueno, Aram Davtyan, Juan J. de Pablo, Peter G. Wolynes.

**Project administration:** Wei Lu, Nicholas P. Schafer, Juan J. de Pablo, Peter G. Wolynes.

**Software:** Wei Lu, Carlos Bueno, Nicholas P. Schafer, Joshua Moller, Xun Chen, Mingchen Chen, Xinyu Gu, Aram Davtyan.

**Supervision:** Juan J. de Pablo, Peter G. Wolynes.

**Validation:** Wei Lu.

**Visualization:** Wei Lu, Carlos Bueno.

**Writing – original draft:** Wei Lu, Carlos Bueno, Nicholas P. Schafer.

**Writing – review & editing:** Wei Lu, Carlos Bueno, Nicholas P. Schafer, Joshua Moller, Juan J. de Pablo, Peter G. Wolynes.

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
