## [Decision Letter · Decision Letter 0]

10 Dec 2020

Dear Prof. Wolynes,

Thank you very much for submitting your manuscript "OpenAWSEM with Open3SPN2: a fast, flexible, and accessible framework for large-scale coarse-grained biomolecular simulations" for consideration at PLOS Computational Biology. As with all papers reviewed by the journal, your manuscript was reviewed by members of the editorial board and by several independent reviewers. The reviewers appreciated the attention to an important topic. Based on the reviews, we are likely to accept this manuscript for publication, providing that you modify the manuscript according to the review recommendations.

Sincerely,

Dina Schneidman-Duhovny

Software Editor

PLOS Computational Biology

Dina Schneidman-Duhovny

Software Editor

PLOS Computational Biology

[LINK]

Reviewer's Responses to Questions

**Comments to the Authors:**

Reviewer #1: Authors present an implementation of the AWSEM force-field for proteins and the 3SPN.2 model for DNA to openMM, together with two new potential energy functions. By the implementation and the use of GPU, protein simulations with AWSEM speed up two orders of magnitude, which is very impressive. In addition, taking advantage of the extensibility of openMM, authors implemented two non-trivial energy functions; one for the hybrid use of water- and membrane- environmental contact energies, and the other for many-body covalent bond potential. Both of the new potentials were shown to improve the simulations significantly. Together, I consider this a very impressive and useful software development and thus I recommend its publication after addressing the following minor points.

1) The speed up by openMM/GPU relative to LAMMPS/CPU reaches two order of magnitude for AWSEM, but is limited to less than 10 for 3SPN.2. Also, for a short DNA sequence, the openMM version is slower. Authors should provide some reasonings of this difference, if possible. At least, some discussions must be possible.

2) Related to the above point, can author describe the dominant part of the calculations with openMM/GPU? Is it contact potential for AWSEM? For 3SPN.2, the bottleneck may be the electrostatic interaction.

3) I like the many-body potential function of disulfide bonds. Yet, I am not aware of the motivation to use theta^near function. Does this increase the specificity? Is it really necessary ?

4) A very minor point. In the section: Availability and Future Directions, I do not find any comments on future directions.

Reviewer #2: The authors implemented existing coarse-grained models for protein (AWSEM) and DNA (3SPN2) under the OpenMM framework. The new implementation enabled MD simulations of protein and/or DNA using GPUs, and the simulation speeds were accelerated dramatically as compared to the implementation in LAMMPS, especially for large proteins. In addition, two new potentials were also proposed: one for the interaction between protein and membrane, the other for proteins with multiple disulfide bonds to avoid unphysical clustering of cysteines. The predicted structures were in good agreement with experimental ones in the applications to protein-DNA complexes, proteins partially buried in a membrane and proteins with multiple disulfide bonds. The software will be useful to the community of molecular biophysics.

Here comes the questions and concerns:

1. In the simulation of the protein-DNA complex, the protein and DNA were first pulled apart. How far apart are the protein and DNA?

2. Following the instruction in github, the tutorial for DNA simulation with open3SPN2 can be successfully completed. However, the tutorial for protein-DNA simulation with open3SPN2 cannot be completed and failed with errors. Could the author double check the code of the tutorial (both the one in the SI and the one online)?

3. With the two new potentials, could the authors comment on the potential applications other than the prediction of protein native structures? How about the simulation of dimerization or oligomerization of membrane proteins? How about the potential kinetics and dynamics studies?

4. To facilitate the wide application of the software, a tutorial on how to simulate proteins partially buried in membrane will be helpful.

5. At the end of page 3 in SI, “The parameters are defined in 1” should be “The parameters are defined in Table 1”

6. On page 9 of SI, the table number is not shown properly.

**Have all data underlying the figures and results presented in the manuscript been provided?**

Reviewer #1: Yes

Reviewer #2: Yes

PLOS authors have the option to publish the peer review history of their article (what does this mean?). If published, this will include your full peer review and any attached files.

Reviewer #1: No

Reviewer #2: **Yes: **Wenjin Li
---

## [Editor Report · Decision Letter 1]

9 Jan 2021

Dear Prof. Wolynes,

We are pleased to inform you that your manuscript 'OpenAWSEM with Open3SPN2: a fast, flexible, and accessible framework for large-scale coarse-grained biomolecular simulations' has been provisionally accepted for publication in PLOS Computational Biology.

Best regards,

Dina Schneidman

Software Editor

PLOS Computational Biology

---

## [Editor Report · Acceptance letter]

10 Feb 2021

PCOMPBIOL-D-20-01547R1 

OpenAWSEM with Open3SPN2: a fast, flexible, and accessible framework for large-scale coarse-grained biomolecular simulations

Dear Dr Wolynes,

I am pleased to inform you that your manuscript has been formally accepted for publication in PLOS Computational Biology. Your manuscript is now with our production department and you will be notified of the publication date in due course.

With kind regards,

Alice Ellingham
